# Thermal Aging Properties of 500 kV AC and DC XLPE Cable Insulation Materials

**DOI:** 10.3390/polym14245400

**Published:** 2022-12-09

**Authors:** Ling Zhang, Zhaowei Wang, Jihuan Tian, Shaoxin Meng, Yuanxiang Zhou

**Affiliations:** 1State Key Laboratory of Power Grid Environmental Protection, China Electric Power Research Institute, Wuhan 430074, China; 2State Key Laboratory of Power System Operation and Control, Department of Electrical Engineering, Tsinghua University, Beijing 100084, China; 3Wuwei Electric Power Supply Company of the State Grid, Wuwei 733000, China

**Keywords:** XLPE, thermal aging, space charge, carbonyl index, antioxidant, trap depth

## Abstract

Despite similar material composition and insulation application, the alternating current (AC) cross-linked polyethylene (XLPE) and direct current (DC) XLPE materials cannot replace each other due to different voltage forms. Herein, this work presents a systematical investigation into the effects of thermal aging on the material composition and properties of 500 kV-level commercial AC XLPE and DC XLPE materials. A higher content of antioxidants in the AC XLPE than in the DC XLPE was experimentally demonstrated via thermal analysis technologies, such as oxidation-induced time and oxidation-induced temperature. Retarded thermal oxidation and suppression of space charge effects were observed in thermally aged AC XLPE samples. On the other hand, the carbonyl index of DC XLPE dramatically rose when thermal aging was up to 168 h. The newly generated oxygen-containing groups provided deep trapping sites (~0.95 eV) for space charges and caused severe electric field distortion (120%) under −50 kV/mm at room temperature in the aged DC XLPE samples. For the unaged XLPE materials, the positive space charge packets were attributed to the residue crosslinking byproducts, even after being treated in vacuum at 70 °C for 24 h. Thus, it was reasoned that the DC XLPE material had a lower crosslinking degree to guarantee fewer crosslinking byproducts. This work offers a simple but accurate method for evaluating thermal oxidation resistance and space charge properties crucial for developing high-performance HVDC cable insulation materials.

## 1. Introduction

The high voltage direct current (HVDC) transmission system plays an important role in obtaining the goal of “carbon peaking and carbon neutrality” under the background of “new infrastructure” in China [1]. HVDC power cable technology is critical to support the attainment of new-energy power transmission [2]. Since 2013, the voltage level of China’s HVDC cross-linked polyethylene (XLPE) insulated cables has increased from ±160 kV to ±535 kV [3]. Current research attention has been paid to developing ±800 kV HVDC XLPE insulated power cables. Although experience in running HVDC XLPE cable projects has been accumulated in China [4], the DC XLPE material still depends on imports. The research, production, test, and operation standards of DC XLPE materials are still immature in China.

The development of insulation material is the key to the technical progress of HVDC cable transmission. Currently, there are three major research approaches: improving purity, grafting modification [5], and nanocomposite [6,7]. The high-purity XLPE materials of Nordic Chemical and Dow Chemical have excellent DC insulation performance but are prone to homocharge injection [8]. The higher the voltage level, the stricter the requirements for impurity size in XLPE samples, which causes greater difficulties in all aspects, such as masterbatch, transportation, preservation, and production [9]. For the nanocomposite approach, the charge injection can be largely inhibited, but the dissipation of space charge becomes difficult at the same time [10]. Furthermore, the interaction between surface-modified nanoparticles and high-molecule polymer matrix is relatively complicated. The dispersion of nanoparticles is governed by a balance of surface energy between the nanoparticles and matrix and, more importantly, the nanoparticle core/core attraction derived by the van der Waals force (vdW) [11,12]. The oxidation resistance and insulation properties of XLPE materials can be flexibly controlled by the grafting modification of antioxidants [13] or voltage stabilizers [14]. However, a problem remains in introducing impurities during the grafting modification process. Enormous studies have been conducted on the space charge and degassing characteristics of XLPE cable materials [15]. In recent years, Chinese cable manufacturers have tried to develop DC XLPE cable materials through different approaches and gained valuable experience in the ±535 kV DC cable tests [16]. However, large-scale commercial-grade HVDC XLPE cable materials have not yet been implemented. It is urgently needed to develop XLPE materials with stable and reliable quality with independent intellectual property rights, to provide technical and equipment support for HVDC transmission projects.

The short-term overheating problem of up to 250 °C, due to overload or short-circuit, is critical to power cables, especially for 500 kV-level HVAC or HVDC cables with an insulation thickness of more than 30 mm [17,18]. The problem mentioned above shows that heat dissipation within the thick cable insulation is a great challenge, which might contribute to the accelerated thermal aging in XLPE insulation performance, including the thermal, mechanical, and electrical properties [19]. To the best of our knowledge, however, there is no direct attention to comparative studies on the thermal aging characteristics of AC XLPE and DC XLPE materials at the 500 kV voltage level. For the engineering application of DC XLPE materials, any change of antioxidant content under operating conditions may be a significant indicator for better understanding the aging behavior of cable insulation materials.

In this work, the physical, thermal, mechanical, and space charge characteristics of imported 500 kV-level commercial AC and DC XLPE cable materials were systematically studied. Thermal analysis measurements and oven aging tests were designed and carried out for the oxidation evaluation and thermal-oxidative aging of AC and DC XLPE materials. The crosslinking byproducts, antioxidant content, and oxidation resistance of the XLPE materials were evaluated. Furthermore, the effects of oxidation reaction products on the space charge, electric field distortion, and trap depth characteristics were investigated to provide guidance for practical HVDC XLPE cable projects.

## 2. Materials and Methods

### 2.1. Materials and Sample Preparation

In this work, two types of widely used 500 kV-level AC and DC XLPE cable material pellets were selected, which were imported from Europe and kindly supplied by Ningbo Orient Wires & Cables Co. Ltd., (Ningbo, China).

The molding and crosslinking processes of XLPE samples were divided into four stages, including hot-pressing molding, high-temperature crosslinking, iso-pressure cooling, and vacuum degassing, to obtain thin films with uniform thickness and no defects such as bubbles. Specifically, the XLPE cable material pellets, covered with polyester films, were dispersed between preheated steel molds for pre-melting at 125 °C for 3 min by hot pressing at 10 MPa to obtain uncrosslinked films with uniform thickness. These films were subsequently cross-linked at 185 °C for 10 min by hot pressing at 15 MPa. Then, the crosslinked films were rapidly cooled to 60 °C while maintaining the 15 MPa pressure to avoid tiny bubbles derived from the crosslinking byproducts. All XLPE film samples were vacuum degassed at 70 °C for 24 h to eliminate the byproducts as much as possible [20].

### 2.2. Methods

#### 2.2.1. Calculation of the Crosslinking Degree

The crosslinking degree of XLPE films was measured according to the Standard ISO 10147:1994. Specifically, samples about 0.50 g were wrapped in a copper net and then immersed into xylene solvent in a round-bottom flask connected to a reflux condenser. Then, the solvent was heated to its boiling point, and the samples were extracted for 24 h under reflux condensation. Then, extracted samples were dried to a constant weight to obtain the crosslinking degree *η* according to Equation (1) [21].
(1)$$\eta =\frac{m_{1}}{m_{2}}\times 100\%$$
where *m*_1_ represents the final mass, mg; and *m*_2_ represents the initial mass, mg.

#### 2.2.2. Mechanical Tests

According to Standard IEC 60811-1-1:2001, the thermal elongation properties of the XLPE samples were tested. Firstly, the dumbbell-shaped XLPE samples were preheated at 200 °C for 10 min, then stretched with a load of 0.20 MPa for 5 min stretching. Then, the high-temperature tensile length and the permanent length were recorded after cooling to room temperature, representing the load elongation and permanent elongation, respectively.

According to Standard IEC 60811-401, XLPE films were thermally aged in an oven with a thermal aging time of up to 168 h. In the preliminary test stage, we found that the 135 °C specified in Standard IEC 60811-401 caused the XLPE films to quickly curl and deform. Thus, 125 °C was eventually chosen as the accelerating thermal aging temperature.

Tensile tests of the thermally aged AC and DC XLPE samples were carried out via an MIT-05 universal testing machine. According to Standard ISO 527-2:1993, the dumbbell-shaped XLPE samples had a thickness of 2 ± 0.2 mm, an effective width of 4 ± 0.1 mm, and a gauge distance of 50 ± 0.5 mm.

#### 2.2.3. Thermal Analysis

Differential scanning calorimetry (DSC) on a TA DSC250 (TA Instruments, New Castle, DE, USA) platform was used to characterize the crystallization and melting properties, as well as the oxidation-induced time and oxidation-induced temperature.

For crystallization and melting analyses, the heat flow was recorded between 0 °C and 200 °C, and circulated for two rounds to eliminate the thermal history with a ramp rate of 20 °C/min under N_2_ flow.

To compare the oxidation properties of AC and DC XLPE samples, oxidation-induced time (denoted as isothermal OIT) was used to measure the oxidation exothermic time of XLPE materials, and oxidation-induced temperature (denoted as dynamic OIT) was used to measure the oxidation exothermic temperature. Both isothermal OIT and dynamic OIT tests complied with the Standard ISO 11357-6:2008.

Thermogravimetric analysis (TGA) was completed to evaluate the thermal-oxidative stability of the AC and DC XLPE samples on a TA TGA55 platform (TA Instruments, New Castle, DE, USA). Specifically, ~8.0 mg XLPE slices were heated from room temperature to 700 °C with a ramp rate of 20 °C/min under O_2_ flow.

#### 2.2.4. FTIR Method

Fourier transform infrared (FTIR) spectroscopy was used to detect the content change of typical functional groups, such as the carbonyl group, of thermally aged XLPE films (~30 μm) via a Nicolet NEXUS 650 spectrometer (Thermal Fisher Scientific, Waltham, MA, USA). The FTIR spectra were collected with a resolution of 2 cm^−1^ under the transmission mode, and the scan number was set as 8.

#### 2.2.5. XPS Method

X-ray photoelectron spectroscopy (XPS) was utilized to obtain the C1s spectra of AC XLPE samples with an Escalab Xi+ instrument (Thermal Fisher Scientific, Waltham, MA, USA). The experimental conditions were set as follows: 500 μm X-ray beam spot, 15 kV voltage, 10 mA current, Al target monochromatic, and magnetic transmission mode. To prepare the samples for XPS measurements, an AC XLPE film was heated at 250 °C in an N_2_ atmosphere using the TGA equipment.

#### 2.2.6. Space Charge Measurement

The space charge behaviors of the thermally aged AC and DC XLPE films (~220 μm) were measured at room temperature utilizing a pulsed electro-acoustic (PEA) system according to Standard IEC/TS 62758-2012. The applied electric field was −50 kV/mm, and the polarization and depolarization durations were 60 min and 10 min, respectively.

## 3. Results

### 3.1. Crosslinking Properties

#### 3.1.1. Crosslinking Degree

A three-dimensional network of polyethylene molecules was formed via covalent bonding during the high-temperature crosslinking reaction process. The crosslinking degree can be used to characterize the proportion of linear polyethylene molecules involved in the three-dimensional network [22]. Previous literature shows that the operating temperature and mechanical properties of XLPE in long-term service increase with the crosslinking degree [23].

Herein, the crosslinking degrees of the commercial AC and DC XLPE film samples were 80% and 67%, respectively. DSC test results showed that no exothermic phenomenon existed in XLPE film samples when heating up to 220 °C, which confirmed that all the crosslinking agents contained in the XLPE masterbatch particles had been activated to crosslink polyethylene molecules during the hot-pressing process described in Section 2.1. Thus, it is reasonable to think that more crosslinking agents participated in crosslinking in the AC XLPE, which might also lead to more crosslinking byproducts.

#### 3.1.2. Thermal Elongation

The thermal elongation test is a rapid and qualitative test method for measuring the crosslinking degree of XLPE materials. It is an important basis for judging the quality of cable products and manufacturing process parameters. The current thermal elongation standard of XLPE cable insulation is no larger than 175%, and the permanent deformation standard is no larger than 15%.

In this work, the thermal elongation of the dumbbell-shaped AC samples was 125% under a standard 0.2 MPa load at 200 °C, while that of the DC samples exceeded the maximum range (750%) of the thermal elongation system. It can be seen that the AC sample still kept certain tensile strength and good creep resistance at 200 °C. In contrast, the DC sample lost its tensile capability completely to suspend a load. After cooling to room temperature and removing the load, the permanent deformation of the AC sample returned to 0. In contrast, the thermal elongation of the DC sample remained at 18%, which indicated that only the AC samples could meet the thermal elongation standard of XLPE cable insulation.

### 3.2. Thermal Properties

#### 3.2.1. Melt and Crystallization Properties

XLPE is a semicrystalline thermosetting polymer, and the crystalline structure greatly influences its macroscopic properties. The melting and crystallization curves were obtained via DSC tests, as shown in Figure 1, and related parameters are listed in Table 1. The crystallinity was calculated according to Equations (2) and (3) [24].
(2)$$X_{c}=\frac{\Delta H_{m}}{\Delta H_{r}^{0}}$$
(3)$$\Delta H_{m}=\int_{T_{1}}^{T_{2}}\frac{F}{R}dT$$
where Δ*H_m_* represents the melting enthalpy of XLPE samples, J/g; *T*_1_ and *T*_2_ are the lower- and upper-temperature limits of the integral, °C; *F* is the heat flow, W/g; *R* represents the ramp rate, °C/s; and Δ*H_r_^0^* represents the equilibrium enthalpy of the fully crystallized polyethylene, which is about 287.3 J/g [25].

It is worth noting that the AC material had a larger full width at half maximum (*FWHM*) of crystallization (9.8 °C) than the DC material (6.3 °C), which meant that the AC material had a worse uniformity of the crystal size distribution. On the contrary, a concentrated crystallization process contributed to the more uniform grain distribution of the DC material, which objectively promoted the improvement of crystallinity (37.4%).

#### 3.2.2. DSC OIT Properties

Thermal-oxidative degradation is one of the main reasons for the property deterioration of XLPE insulation materials. In the practical engineering application, a small amount of antioxidants is essentially added to polymers to inhibit the oxidative free radicals and peroxides generated during long-term service [26]. The isothermal OIT test results are shown in Figure 2a. The isothermal OIT of the AC material reached 22.70 min, which was much longer than that of the DC material (4.37 min). It was speculated that the AC material contained more antioxidants than the DC material.

The dynamic OIT test results are shown in Figure 2b. The dynamic OIT of the DC and AC materials was 240.4 °C and 248.3 °C, respectively. Thus, it can be confirmed again that fewer antioxidants were added to the DC material. Both isothermal and dynamic OIT results are listed in Table 2. Shimada et al. [27] found that when the antioxidant content was as low as 0.04% (10^−6^ mol/cm^3^), its inhibition effect on the thermal oxidation of XLPE materials would be significantly reduced.

#### 3.2.3. Thermal Oxygen Decomposition

The thermal oxygen decomposition of AC and DC XLPE materials is shown in Figure 3. Specifically, the TGA curves could be divided into four stages according to the weight loss. In stage 1 (room temperature ~240 °C), the residual mass was larger than 99%, indicating that a small amount of water and impurities volatilized. In stage 2 (240~260 °C), the sample reached the thermal-oxidative temperature, and the oxidation reaction was exothermic, as verified by Figure 2b. In stage 3 (260~300 °C), the C−C bond was thermally cracked, and the residual mass decreased rapidly. In stage 4 (300~550 °C), the oxygen-containing groups were thermally broken, and the XLPE samples were completely decomposed.

It is worth noting that both TGA curves in the blue dotted box in Figure 3 showed a temperature decrease (~20 °C). It was speculated that as the temperature rose linearly and approached 300 °C, the thermal cracking rate of the C−C bonds in stage 3 was accelerating (which can be confirmed by the increasing slopes of the two curves in the figure). When the endothermic rate of the C−C bond thermal cracking far exceeded the heat provided by the TGA test system, the local temperature around the samples (also near the temperature sensor) dropped.

After the TGA curve reached stage 2, it entered the thermal-oxidative process, as shown in the partially enlarged diagram in Figure 3. Oxygen molecules consumed the antioxidants more quickly with the linearly increasing temperature. When the antioxidants were exhausted, oxygen molecules reacted with the free radicals (R) on the molecular chain of XLPE, and oxygen-containing free radicals (ROO∙) were generated, which was reflected by a slight increase in the TGA mass curve. With the continuous temperature increase, the thermal-oxidative reaction rate increased, leading to the continuous growth of sample mass. Here, the thermal-oxidative temperature was defined as the intersection of the maximum mass increase rate tangent and the baseline [28].

The 5% weight loss of thermal decomposition could be used to evaluate the thermal aging life of XLPE samples, and the temperature was defined as a 5% weight loss temperature. Table 2 summarizes the thermal oxidation and decomposition parameters of AC and DC XLPE materials. The thermal-oxidative temperature of the DC cable material was 240.2 °C, and the 5% weight loss temperature was 252.8 °C, which were both lower than those of the AC material (242.2 °C and 259.1 °C), indicating that the thermal oxidation and decomposition performances of the AC material were better than those of the DC material.

### 3.3. Space Charge Properties

Polar small-molecule antioxidants in XLPE will contribute to the accumulation of space charges [29]. Due to the negligible space charge effect under AC voltages, more antioxidants were added to the AC XLPE material to improve the oxidation resistance in long-term and high-temperature service, avoiding harmful local charge accumulation [30]. Taking the charge accumulation as a reference, the AC material had more margin for adding antioxidants. By comparison, a smaller amount of antioxidants was added to the DC material, leading to a weakening of the oxidation resistance. This was a comprehensive adjustment and control of material parameters to meet the requirements of engineering applications to achieve the balance between thermal aging and insulation performances.

In addition to antioxidants, previous studies also paid much attention to the effect of crosslinking byproducts on the electrical properties of XLPE materials, especially space charge behaviors. The vacuum treatment at 70 °C for 24 h of XLPE samples was described in Section 2.1 in detail. Nevertheless, it still needs further investigation as to whether crosslinking byproducts can be eliminated via the above-mentioned method.

Figure 4 shows the space charge and electric field distribution of unaged AC and DC XLPE materials within 60 min under −50 kV/mm. In the initial stage of electrical polarization, the AC XLPE sample was injected into positive space charge packets from the anode and then migrated toward the cathode for recombination and precipitation. With continuous polarization, the positive charge around the anode gradually changed to negative charge accumulation, and the charge density gradually increased. The electric field distorted as positive space charge packets migrated, and the maximum distortion ratio of the internal electric field was 69.2%, existing near the anode.

In the initial stage of electric polarization, injection and migration of positive space charge packets also occurred at the anode of the DC XLPE sample. However, the space charge packets disappeared after 10 min of polarization, and only a small amount of positive space charge eventually accumulated and spread out evenly in bulk. The maximum distortion ratio of the internal electric field was 32.6%.

The charge accumulation of the DC sample near the electrode showed the same polarity. The charge behavior of the AC sample changed from homocharge to heterocharge accumulation, and then gradually increased. It was speculated that even after 24 h of vacuum degassing, there were still more crosslinking byproducts and small polar molecules, such as antioxidants, in the AC material than in the DC material, and deep traps near the anode trapped a large number of electrons. Meanwhile, holes migrated to the cathode and precipitated out.

It should be pointed out that the space charge test of the AC material under DC voltages was mainly based on the comparative study. The obtained results cannot explain the space charge characteristics of the AC material under AC voltages in practical engineering applications.

### 3.4. Thermal Aging Properties

#### 3.4.1. Tensile Properties of Thermally Aged XLPE Samples

The elongation at break can represent the deformation capacity of XLPE samples. As shown in Figure 5, the elongation at break of the AC and DC XLPE samples increased after 6 h of thermal aging, while showing a gradual downward trend with continuous thermal aging. Overall, the elongation at break of the DC samples was larger than that of the AC samples.

The breaking strength can represent the maximum tensile strength of XLPE samples. In Figure 5, the fracture strength trend was similar to the elongation at break. The fracture strength of the AC and DC XLPE samples increased after 6 h of thermal aging. In conclusion, the deformation capacity of the DC material was better, but the maximum tensile capacity was not as good as that of the AC material. This was consistent with the trend of the previous crosslinking and thermal elongation test results. The higher the crosslinking degree, the better the maximum tensile strength of XLPE. In addition to crosslinking degree, deformation ability and ultimate tensile strength are also related to the molecular chain structure, molecular weight, and branching degree of XLPE [31].

The thermal aging process of XLPE samples can be divided into physical and chemical aging processes. Before the complete consumption of antioxidants, the sample was in the physical aging process, and the recrystallization process tended to cause the imperfect crystals to grow, and the crystallinity increased. After the antioxidant consumption, the sample was in the chemical aging process, and the oxidation reaction caused the XLPE molecular chain to break, and the crystallinity decreased. At the initial stage of thermal aging, the elongation at break and fracture strength of both XLPE samples increased. This was because the recrystallization process of the samples was affected by high temperatures, and the molecular arrangement became more orderly. Thus, their mechanical properties were enhanced [32]. With the extension of thermal aging time, the surface layer of the dumbbell-shaped XLPE samples was oxidized, the molecular chain was partially cracked, and its elongation at break and breaking strength gradually decreased.

#### 3.4.2. FTIR Spectra of Thermally Aged XLPE Samples

The FTIR spectra of thermally aged AC and DC XLPE materials are shown in Figure 6. The infrared spectral transmittance determined that the carbonyl (C=O) content was close to the wavenumber of 1722 cm^−1^. The transmittance of 168-h-aged DC XLPE samples decreased near 1722 cm^−1^, indicating that the carbonyl content increased [33]. Meanwhile, a broad absorption peak appeared in the wavenumber range of 3200~3500 cm^−1^, corresponding to the hydroxyl group (−OH). However, the FTIR spectra of the AC XLPE film with different thermal aging times showed little change, indicating that it had not yet entered the stage of rapid thermal-oxidative reaction.

#### 3.4.3. Space Charge of Thermally Aged XLPE Samples

Figure 7 shows the space charge and electric field distribution of the thermally aged AC XLPE samples at room temperature polarized under −50 kV/mm for 60 min. It can be seen from Figure 4 that positive space charge packets were injected from the anode into the unaged AC XLPE samples, while no space charge packets were observed in the thermally aged AC XLPE samples. In the early stage of thermal aging, there was a recrystallization process in the sample, making the crystal structure perfect. The antioxidant was gradually consumed, and the residual crosslinking byproducts continued to precipitate and volatilize at high temperatures. Thus, the space charge accumulation gradually decreased.

The negative charge in the AC XLPE sample decreased from the anode to the cathode with thermal aging. After aging for 96 h and 168 h, only a tiny amount of negative space charge accumulated near the anode. At this time, thermal cracking dominated the thermal-oxidative aging process. XLPE underwent an oxidation reaction, breaking macromolecular chains and causing the appearance of oxidation products containing unsaturated groups. Therefore, a small number of heterocharges accumulated near the anode. As the AC XLPE material contained more antioxidants, the oxidation reaction only occurred locally. The change of space charge distribution characteristics was not apparent, even when aged up to 168 h.

Figure 8 shows the space charge and electric field distribution of the DC XLPE samples at room temperature polarized under −50 kV/mm for 60 min. It can be seen from Figure 4 that the unaged DC XLPE sample had a positive space charge packet injection at the initial stage of polarization. The 24-h-aged sample only had a small amount of positive charge injection, and the 96-h-aged sample had no positive charge accumulation. In comparison, the 168 h-aged samples accumulated many negative space charges near the anode.

It is worth noting that the charge accumulation behaviors at the anode of the DC and AC XLPE samples were similar until 96 h of thermal aging. With the continuous thermal aging, the homocharge accumulation near the anode changed to heterocharge accumulation, which might reflect the antioxidant consumption and the continuous precipitation of crosslinking byproducts.

According to the FTIR spectra in Figure 6, the carbonyl and hydroxyl groups of the 168-h-aged DC XLPE sample increased rapidly, indicating that the antioxidant was exhausted at that time, and a strong thermal-oxidative reaction occurred at 125 °C. The macromolecular chain broke, and the appearance of a large number of oxidative free radicals and polar groups introduced many newly generated deep traps, which promoted the heterocharge accumulation near the anode.

## 4. Discussion

The crosslinking degree and DSC crosslinking exothermic results indicated that the AC material might produce more crosslinking byproducts than the DC material. Even after degassing in vacuum at 70 °C for 24 h, the crosslinking byproducts could not be entirely removed, which would directly impact the space charge behaviors of the unaged and thermally aged XLPE films. This work adopted TGA treatment and XPS spectrum to confirm the objective occurrence of residue byproducts.

Firstly, TGA equipment was utilized to heat-treat AC XLPE film at 250 °C in an N_2_ atmosphere for 100 min to accelerate the removal of residual byproducts in the film in a short time. The related parameters were set for the following reasons. N_2_ atmosphere can guarantee no appearance of thermolysis effect at 250 °C. Moreover, the heat-treatment temperature of 250 °C also refers to the highest temperature of short-term overheating in a practical cable project [33]. During the TGA heat treatment for 100 min, the residue mass gradually decreased and then reached saturation, indicating that 100 min was sufficient.

Then, the XPS C1s spectrum was adopted to characterize the heat-treated AC XLPE sample. Figure 9 shows that the AC XLPE sample had a small peak at 288.0 eV, corresponding to the carbonyl group (C=O). The AC sample was not subjected to thermal-oxidative aging, and the existence of carbonyl groups could have been derived from the crosslinking byproduct acetophenone. It showed that acetophenone still existed even after 250 °C N_2_ heat treatment for 100 min, which proved that degassing in vacuum could not wholly remove byproducts. In addition, the 285.6 eV characteristic peak corresponded to C−OH, which might have been derived from antioxidant molecules, and the 284.8 eV characteristic peak corresponded to the C−C backbone of the AC XLPE samples.

The deterioration of the aggregated structure and properties of XLPE samples was due to thermal-oxidative degradation, in which the free radicals were generated in the XLPE sample (See Figure 10). The free radicals further cracked XLPE molecules under the combination of high temperature and oxygen, which caused the performance of XLPE to continuously deteriorate. After the C−C bonds in the main chain broke, lone electron pairs on C and O recombined to form ketocarbonyl (C=O).

The carbonyl index can be used to reflect and analyze polymers’ thermal-oxidative aging resistance. The results showed that the increase in carbonyl content led to the rise of deep trap density in the sample [34]. As shown in Figure 11 and Table 3, the carbonyl indexes of AC and DC materials were similar and kept relatively stable (1.60 ± 0.2) when the aging time was less than 96 h. However, during the thermal aging process from 96 h to 168 h, the carbonyl index of the DC material dramatically increased from 1.65 to 9.13, which showed that the antioxidants inside the 96-h-aged sample had been exhausted, and the DC XLPE sample underwent a violent oxidation reaction at 125 °C. Figure 11 shows the thermal-oxidative aging resistance of the AC material was better than that of the DC material.

Figure 12 illustrates the schematic diagrams of the thermal-oxidative aging process in the XLPE samples before and after the exhaustion of antioxidants (taking hindered phenol antioxidants, for example). As shown in Figure 12a, the phenolic hydroxyl H atoms in the hindered phenol antioxidants (Ar−OH) were easily lost and captured by free radicals such as R∙, ROO∙, RO∙, and ∙OH, resulting in products such as RH, ROOH (further decomposed into RO∙ and ∙OH), ROH, and H_2_O, and stopping the continuous oxidation of XLPE chains [35]. However, with the extension of thermal-oxidative aging, the hindered phenol antioxidants were gradually exhausted and the thermal oxidation process occurred as shown in Figure 12b, including initiation of free radicals, growth of oxidized products (propagation), and termination of free radicals with the combination of the high temperature and O_2_ atmosphere.

According to the space charge and electric field distribution of XLPE samples shown in Figure 4, Figure 7 and Figure 8 polarized under −50 kV/mm for 60 min, the maximum electric field distortion of XLPE samples with different thermal aging times was obtained, as shown in Figure 13 and Table 3. For the AC material, the maximum electric field distortion decreased at first and then tended to be stable below 20%, which corresponded to a small amount of accumulated charge. For the DC material, the maximum electric field distortion also decreased to 12.4% when aging for 24 h. In contrast to the AC material, it then increased at an accelerating rate, up to 120.0%, due to the large amount of heterocharges near the anode. It can be seen from Figure 4 that the maximum electric field distortion of the unaged AC material was twice that of the DC material, indicating that the unaged AC XLPE material more easily accumulated space charges due to more crosslinking byproducts and antioxidants. When applied to power cable insulation, more positive and negative charges accumulate and form severe electric field distortion inside the insulation for a long time, accelerating the aging of insulation materials.

Considering the severe electric field distortion in the thermally aged DC XLPE samples, it is essential to investigate the influence of thermal-oxidative aging on the trap parameters. The average trap depth of the thermally aged DC XLPE material can be calculated from the space charge data obtained in the depolarization process shown in Figure 14 [36]. The average trap depth of the unaged DC XLPE sample was the smallest due to the low impurity content. After 24 h and 96 h of thermal aging, the contents of antioxidants and crosslinking byproducts decreased continuously due to consumption or volatilization, the recrystallization process of the sample was more extensive than oxidative decomposition-n, and the average trap depth gradually increased. The average trap depth of the 168-h-aged sample increased significantly, and the antioxidant was exhausted, resulting in many polar groups (such as carbonyl groups) acting as deep traps. Therefore, the distribution characteristics of the space charge and electric field deteriorated rapidly.

## 5. Conclusions

In summary, two kinds of 500 kV-level commercial AC and DC XLPE cable materials were selected to systematically investigate the thermal aging effects on the material composition and space charge behaviors. This work focused on two species, i.e., the antioxidants and the crosslinking byproducts, to determine their roles in the thermal-oxidative process and charge transport.

The isothermal OIT, dynamic OIT, thermal-oxidative temperature and 5% thermal weight loss temperature of unaged DC XLPE material were lower than those of unaged AC material, owing to a smaller antioxidant content in the DC material. However, a quantitative correlation between the antioxidant content and the thermal oxidation resistance remains elusive. The carbonyl index of the DC material increased exponentially from 96 h to 168 h of aging at 125°C in the oven. Concurrently, the distribution characteristics of space charge and electric field deteriorated rapidly. Furthermore, the calculated average trap depth from the measured space charge profiles of the 168 h thermally aged DC XLPE sample was much higher than that of the unaged DC XLPE sample. The large discrepancy of the average trap depth was attributed to the exhausted antioxidants and newly-generated oxygen-containing groups. The nature of these oxygen-containing groups and their effects on the intrinsic trap states are still not well understood, which undoubtedly brings more uncertainty for the design and engineering applications of DC XLPE insulation materials.

It is also worth noting that crosslinking byproducts were another important factor that led to degradation of the electrical properties, especially space charge accumulation and internal electric field distortion. XPS analysis proved that the XLPE samples treated at 250 °C for 100 min still contained residual crosslinking byproducts (C=O in the acetophenone molecules). This result reasonably explained the phenomena of space charge packets in both unaged AC and DC XLPE samples after degassing in vacuum at 70 °C for 24 h. However, further efforts focusing on the relationship between the byproduct content and thermal aging are encouraged. In the future, we plan to optimize the additive strategy of antioxidants and reduce the influence of crosslinking byproducts to synergistically improve the comprehensive properties of DC material, including thermal aging properties.

## Figures and Tables

**Figure 1 polymers-14-05400-f001:**
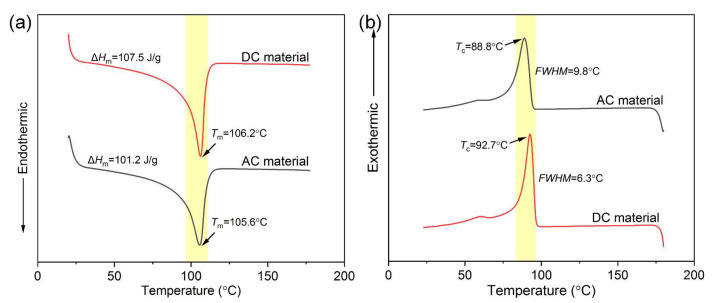
(**a**) Melting and (**b**) crystallization curves of AC and DC XLPE materials.

**Figure 2 polymers-14-05400-f002:**
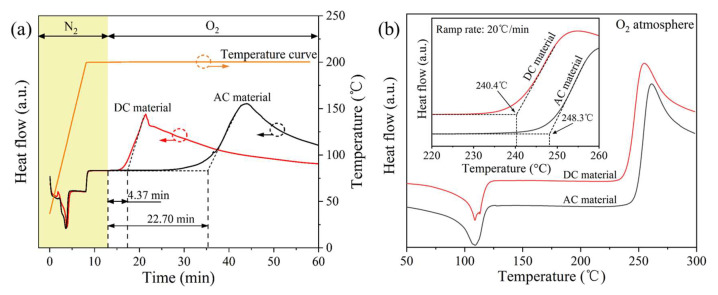
(**a**) Isothermal OIT and (**b**) dynamic OIT, of AC and DC XLPE cable materials.

**Figure 3 polymers-14-05400-f003:**
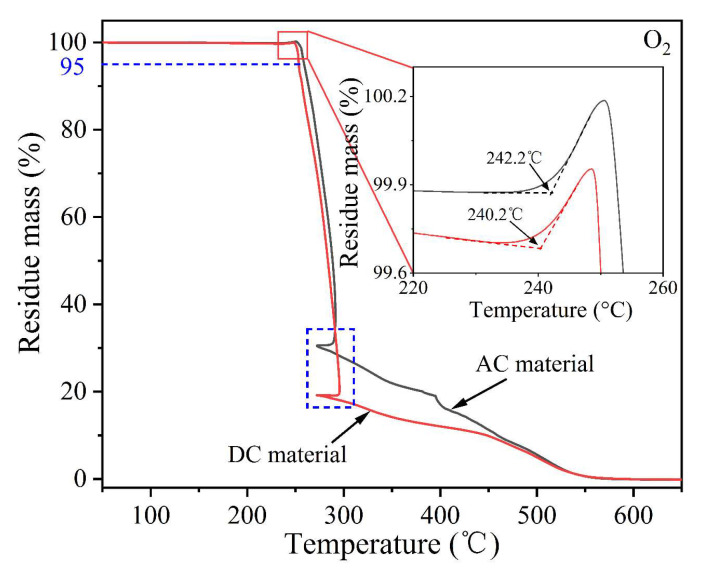
TGA curves of AC and DC XLPE materials in an O_2_ flow.

**Figure 4 polymers-14-05400-f004:**
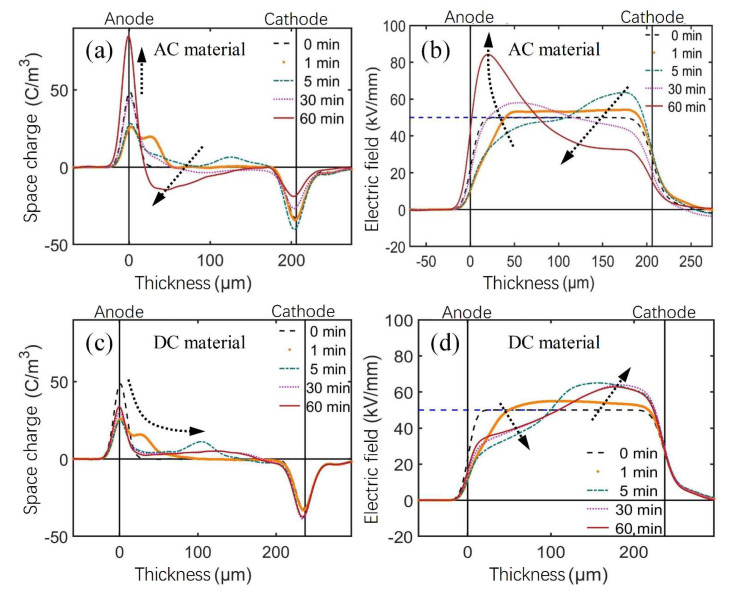
(**a**,**c**) Space charge, (**b**,**d**) electric field distribution, of AC and DC materials polarized under −50 kV/mm for 60 min at room temperature. (Arrows show the variation tendency.)

**Figure 5 polymers-14-05400-f005:**
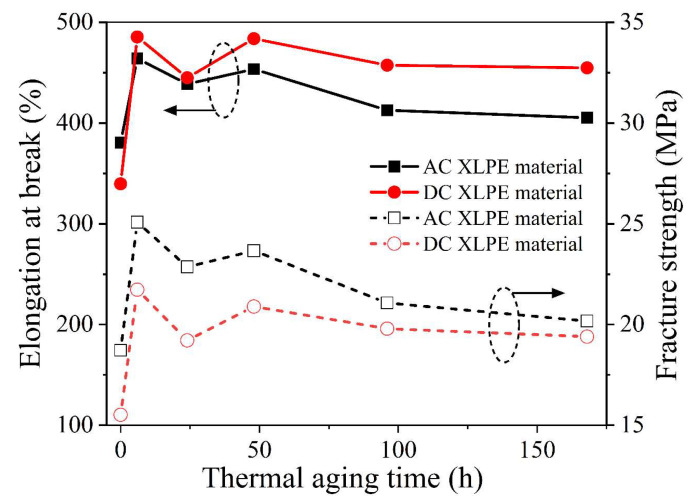
Tensile properties of XLPE materials thermally aged in a 125 °C oven. (Arrows indicate the attribution of the curves at the corresponding ordinate).

**Figure 6 polymers-14-05400-f006:**
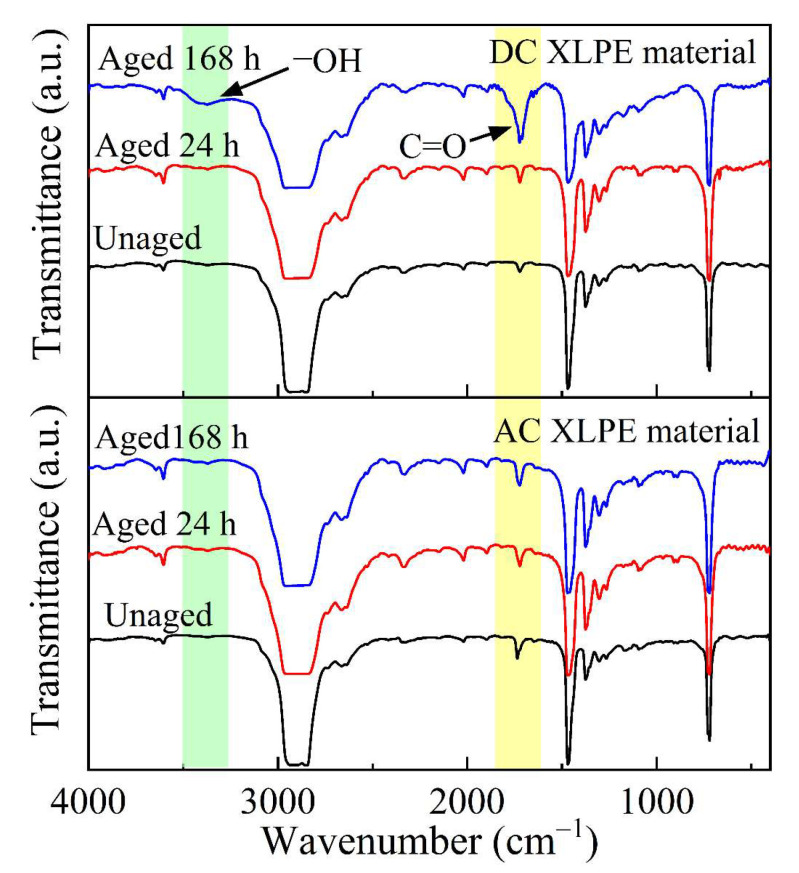
FTIR spectra of AC and DC XLPE materials. (Arrows show the correspondence between functional groups and the wavenumber.)

**Figure 7 polymers-14-05400-f007:**
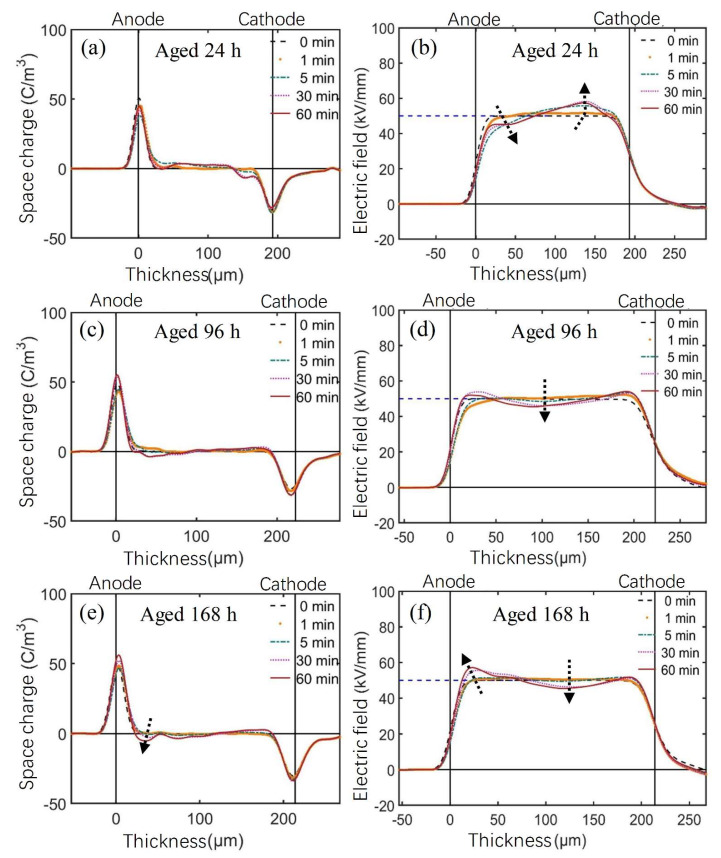
(**a**,**c**,**e**) Space charge, and (**b**,**d**,**f**) electric field distribution, of thermally aged AC XLPE materials polarized under −50 kV/mm for 60 min at room temperature. (Arrows show the variation tendency.)

**Figure 8 polymers-14-05400-f008:**
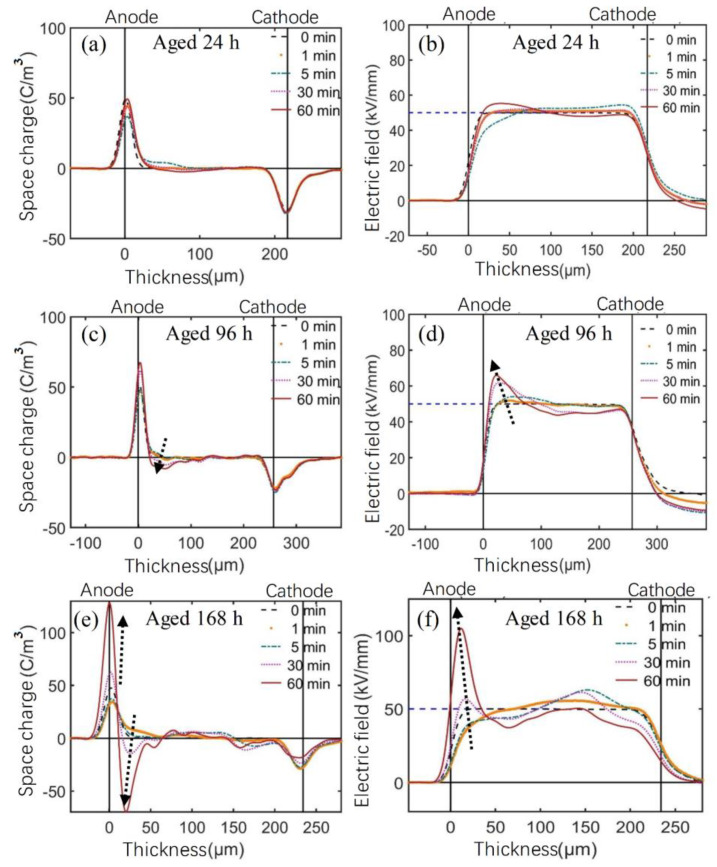
(**a**,**c**,**e**) Space charge, and (**b**,**d**,**f**) electric field distribution, of thermally aged DC XLPE material polarized under −50 kV/mm for 60 min at room temperature. (Arrows show the variation tendency.)

**Figure 9 polymers-14-05400-f009:**
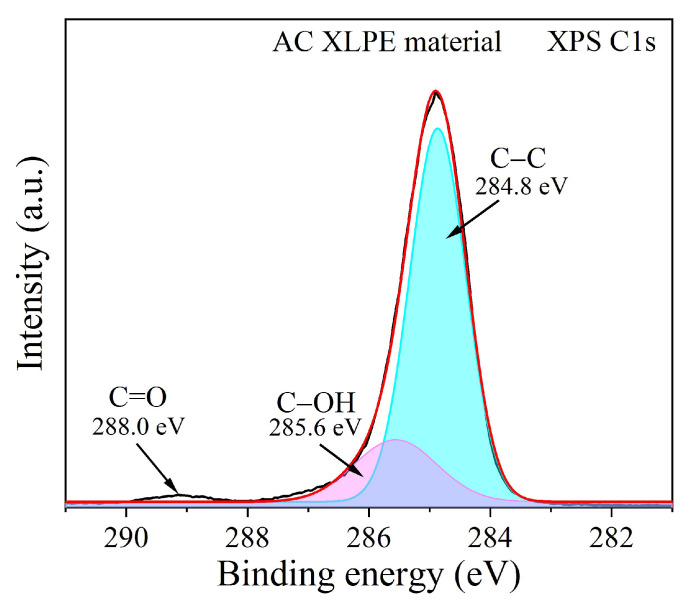
XPS C1s spectrum of AC XLPE material was treated with 250 °C high-temperature N_2_ atmosphere in advance.

**Figure 10 polymers-14-05400-f010:**
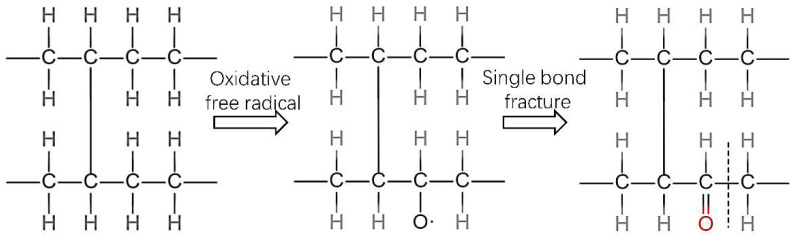
One way to produce a carbonyl group.

**Figure 11 polymers-14-05400-f011:**
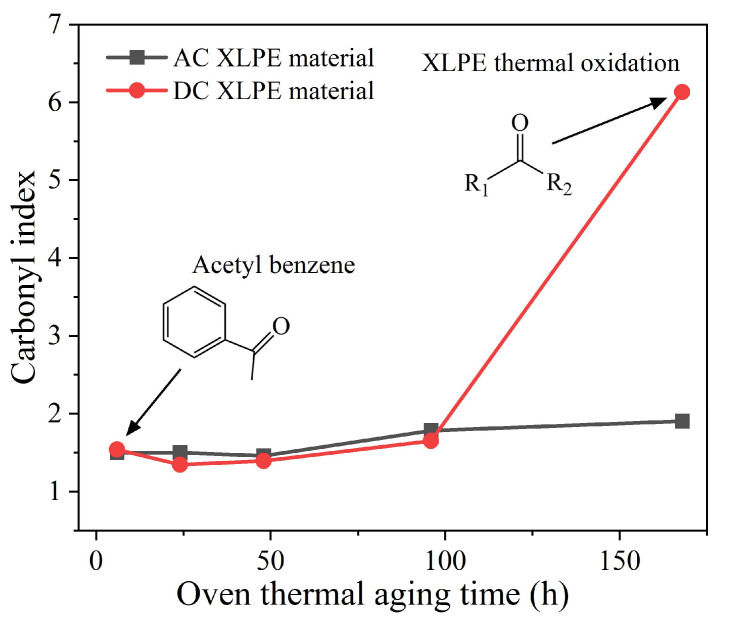
Carbonyl index of thermally aged AC and DC XLPE materials.

**Figure 12 polymers-14-05400-f012:**
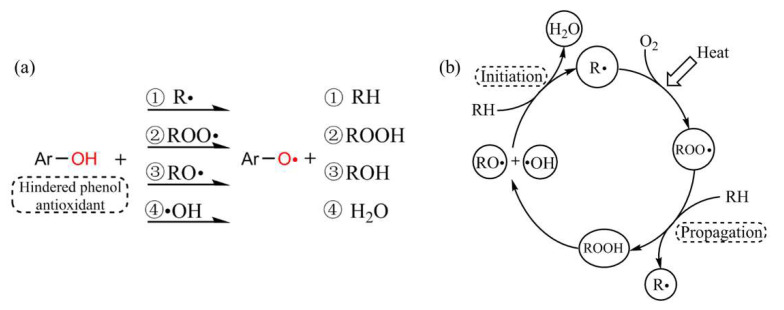
Schematic diagrams: (**a**) hindered phenol antioxidants capture the free radicals and suppress the thermal aging process, (**b**) initiation and propagation of thermal oxidation of XLPE chains after the exhaustion of antioxidants under the combination of high temperature and O_2_ atmosphere.

**Figure 13 polymers-14-05400-f013:**
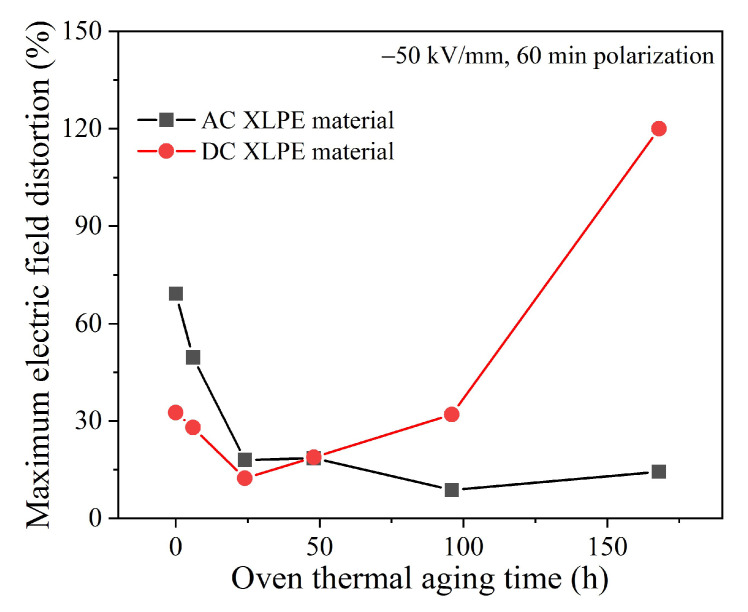
Maximum internal field distortion of thermally aged XLPE materials.

**Figure 14 polymers-14-05400-f014:**
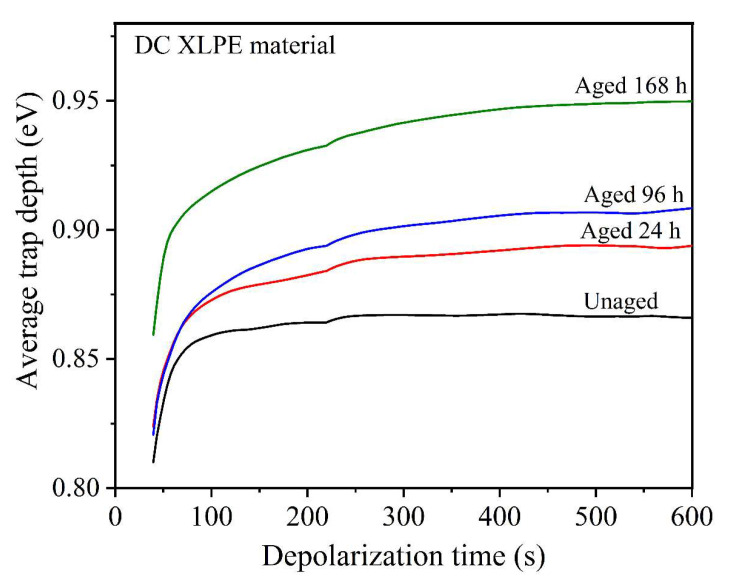
Average trap depth of thermally aged DC XLPE material in the depolarization process.

**Table 1 polymers-14-05400-t001:** Melting and crystallization properties of AC and DC XLPE materials.

	AC Material	DC Material
Melting point *T_m_* (°C)	105.6	106.2
Melting enthalpy Δ*H_m_* (J/g)	101.2	107.5
Crystallinity *X_c_* (%)	35.2	37.4
Crystallization point *T_c_* (°C)	88.8	92.7
*FWHM* (°C)	9.8	6.3

**Table 2 polymers-14-05400-t002:** Thermal oxidation and decomposition properties of XLPE cable materials.

Apparatus	Parameters	AC Material	DC Material
DSC	Isothermal OIT (min)	22.70	4.37
DSC	Dynamic OIT (°C)	248.3	240.4
TGA	Thermal-oxidative temperature (°C)	242.2	240.2
TGA	5% weight loss temperature (°C)	259.1	252.8

**Table 3 polymers-14-05400-t003:** Carbonyl index and maximum internal field distortion of aged XLPE materials.

Parameters	Thermal Aging Time	AC Material	DC Material
Carbonyl index	6 h	1.50	1.54
	24 h	1.47	1.34
	48 h	1.46	1.39
	96 h	1.78	1.65
	168 h	1.90	6.13
Maximum internal field distortion (%)	6 h	49.6	28.0
	24 h	18.0	12.4
	48 h	18.6	18.8
	96 h	8.8	32.0
	168 h	14.4	120.0

## Data Availability

The data presented in this study are available on request from the corresponding author.

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
