# Peer review of "Thermal Aging Properties of 500 kV AC and DC XLPE Cable Insulation Materials"

_polymers, 2022, doi:10.3390/polym14245400_

Round 1
Reviewer 1 Report
The study is interesting and can be published in the Journal after major modifications.
The introduction is sufficient, but it requires more references,
No need to add such data, like lines 73-74 "Organic additives, such as crosslinking agents and antioxidants, are contained inside the XLPE pellets, but the specific data are not disclosed". However, add the name of the cooperative organization that supplied the resin.
Authors must specify that why these temperatures and durations were selected for the thermal aging. Better to add the discussion part in the related sections, like the Antioxidant Performance section is suitable for the TGA analysis.
Add the possible chemical scheme for the rise in the wt. lines 191-200, for a better understanding of the readers.
Modify the Abstract and conclusion.
Cross-check the grammar and spelling throughout the manuscript.
Reviewer 2 Report
The manuscript written on the given topic i.e. Thermal Aging Properties of 500 kV AC and DC XLPE Cable 2 Insulation Materials, is of the great interest for the readers and science hub, but there are some serious scientific as well as language errors in the overall quality of manuscript based on that I have some specific comments
1. Introduction is very limited and at some points more self citations, i would recommend to extend it and globally cite the relevant work
2. Material and Method is hard to understand and can be explained in a better way, in a more detailed way by adding sub heading of each section,
3. Results are too confusing and hard to related with the methods used, therefore I would strongly recommend this section must be carefully revised by adding more tabular and statistical presentation
4. Add tables with the existing figures
5. I would recommend to write discussion in a paragraph form rather than by writing it in points, make a logical flow with the results of existing data and must add future prospects of the work and its significance
6. Conclusion must be revised accordingly as stated above
Round 2
Reviewer 1 Report
After modification acceptance is suggested
Reviewer 2 Report
The manuscript has been modified well, I recommend to accept it in current form